# Isolation and Identification of *Bacillus subtilis* LY-1 and Its Antifungal and Growth-Promoting Effects

**DOI:** 10.3390/plants12244158

**Published:** 2023-12-14

**Authors:** Ying Li, Xia Zhang, Kang He, Xinying Song, Jing Yu, Zhiqing Guo, Manlin Xu

**Affiliations:** 1Shandong Peanut Research Institute, Qingdao 266100, China; li1989ying0921@163.com (Y.L.); zhangxia2259@126.com (X.Z.); sdauhk@163.com (K.H.); songxinying88@126.com (X.S.); iamyujing2008@126.com (J.Y.); zhiqingivy2011@hotmail.com (Z.G.); 2National Engineering Research Center for Peanut, Qingdao 266100, China

**Keywords:** *Bacillus subtilis*, *Fusarium* spp., antagonistic activity, growth-promoting, biological control

## Abstract

Peanut root rot, caused by *Fusarium* spp., is a devastating fungal disease. As part of a program to obtain a biocontrol agent to control peanut root rot in the field, a bacterial strain LY-1 capable of inhibiting the growth of the fungus in vitro was isolated from rhizosphere soil samples collected from wild mint by agar disk dilution and dual-culture assay. Strain LY-1 was identified as *Bacillus subtilis* based on morphological characteristics, 16S rDNA, and *gyrA* sequence analyses. The bacterial suspension and cell-free culture filtrate of LY-1 could significantly inhibit the growth of *Fusarium oxysporum*, *Fusarium proliferatum* and *Fusarium solani*, but volatile organic compounds from the cultures had only a weak effect on mycelial growth. The percentage inhibition of 20% concentration of the cell-free culture filtrate of LY-1 on conidium production of each of the three *Fusarium* species was greater than 72.38%, and the percentage inhibition by the culture filtration on the germination of conidia of the three species was at least 62.37%. The production of extracellular enzyme activity by LY-1 was studied in functional assays, showing protease, cellulase, amylase, chitinase, and β-1,3-glucanase activity, while LY-1 contained a gene encoding iturin, an antifungal lipopeptide. In addition, under pot culture in a greenhouse, culture filtrate of LY-1 significantly promoted the growth of peanut, increasing the fresh and dry mass of the plant by 30.77% and 27.27%, respectively, in comparison with the no-filtrate control. The culture filtrate of LY-1 increased the resistance of peanut plants to *F. oxysporum*, with the biocontrol efficiency reaching 44.71%. In conclusion, *B. subtilis* LY-1, a plant-growth-promoting rhizobacterium, was able to protect peanuts from *Fusarium* spp. infection.

## 1. Introduction

Peanut (*Arachis hypogaea* L.) is an economically important high-protein and high-oil crop in China and plays an important role in ensuring domestic food oil supply and satisfying the diversification of people’s food sources. At present, the peanut planting area in China reaches more than 4.67 million hm², ranking first in the world [1,2]. As a dominant crop in global trade, peanut plays an indispensable role in increasing farmers’ income and foreign trade. However, during the cultivation of peanut plants, the presence of fungal pathogens is a major threat, as these pathogens can affect the quality and quantity of peanut production. Peanut root rot, caused by *Fusarium* spp., is a significant disease that is considered to be one of the major causes of peanut economic losses worldwide [3]. Among the fungi responsible for peanut root rot disease are *Fusarium oxysporum*, *Fusarium proliferatum*, and *Fusarium solani*. Peanut roots are not only infected by *Fusarium* mycelium, but the large number of conidia produced by *Fusarium* spp. can directly invade the epidermis, especially through wounds, of the host plant [4]. 

At present, the control of peanut root rot relies heavily on fungicides, but the large-scale use of such chemicals seriously affects the health of humans and the soil [5]. The most cost-effective control method is the breeding of disease-resistant varieties. However, this method takes a long time, while the adaptive evolution of virulent races of pathogenic microorganisms able to overcome monogenic disease resistance undermines the disease-resistant plant’s ability to control the pathogen [6]. So far, no plant materials with resistance to *Fusarium* root rot have been identified. Therefore, biological control, as a safe and environmentally friendly control method, has attracted considerable attention, such as the application of plant rhizosphere, phyllosphere, or endophytic bacteria, as well as the use of plant-growth-promoting bacteria (PGPB) with antagonistic activity against pathogens [7,8]. 

It has been reported that the biocontrol bacteria and fungi that have been shown to control plant root rot caused by *Fusarium* mainly include species of *Bacillus* [9], *Penicillium* [10], *Pseudomonas* [11], *Streptomyces* [12], and *Trichoderma* [13]. Among these, *Bacillus* has several beneficial characteristics, such as UV resistance, salt tolerance, and heat tolerance [14,15,16]. Because several *Bacillus* spp. have multi-layered cell wall structures and can produce long-lived endospores, they can survive in the natural environment for a long time after introduction and have been developed as commercial bio-fungicides [17]. *Bacillus* spp. play an important role in the biological control of plant diseases and are widely used in agricultural crop production. *Bacillus* spp. can produce two types of antagonistic factors that have antimicrobial effects on plant pathogens, including lipopeptide antifungals and antagonistic proteins [18]. *Pseudomonas* and *Bacillus* species have been the most studied agents in the biocontrol of *Fusarium*, operating through a variety of mechanisms, such as siderophore-mediated iron competition, antifungal metabolite production, and induced systemic resistance [19].

In the current study, we characterized and identified a new antifungal *Bacillus subtilis* strain isolated from the rhizosphere of wild mint, studied its inhibitory effect on the mycelial growth of three *Fusarium* species that can cause root rot in peanut, analyzed the mechanism of inhibiting pathogenic fungi (antifungal peptides and extracellular enzymes), and investigated the potential use of the *B. subtilis* strain as a bioinoculant to enhance peanut growth directly and protect peanut seedlings against *F. oxysporum*.

## 2. Results

### 2.1. Isolation and Identification of Biocontrol Bacterial Strain LY-1

A biocontrol bacterial strain, LY-1, with clear antagonistic effects against *Fusarium* root rot of peanut, was isolated by screening by agar disk dilution and dual-culture assay. The bacterial colony is milky white, round, with irregular edges, wet, opaque, and with a wrinkled surface (Appendix A). It was preliminarily determined that the strain belonged to the genus *Bacillus*. The effective lengths of 16S rDNA and *gyrA* gene sequences from LY-1 were 1391 bp and 783 bp, respectively. Based on the 16S rDNA (GenBank access No. OP199048) and *gyrA* gene sequences (GenBank access No. OP221239), the results of a phylogenetic tree (Figure 1) showed that LY-1 was in the same branch as *Bacillus subtilis* NBRC 13719, with the sequence similarity to *B. subtilis* of 16S rDNA and *gyrA* genes reaching 99.57 and 99.19%, respectively. According to the morphological characteristics of LY-1 and the 16S rDNA and *gyrA* gene sequence analyses, LY-1 was identified as *Bacillus subtilis*.

### 2.2. Antagonistic Activity of B. subtilis LY-1 against Peanut Fusarium *spp*.

As shown in Figure 2, the strain LY-1 demonstrated antagonistic activity against *Fusarium* spp., namely *F. oxysporum*, *F. proliferatum,* and *F. solani*, in dual cultures compared with the control plates; the bacterial suspension of strain LY-1 significantly suppressed the hyphal growth of these three fungal species, *Fusarium* spp., and the percentage inhibition values were 78.61%, 73.13%, and 80.28%, respectively (Figure 2C).

To test for any inhibitory effect of the cell-free culture filtrate of strain LY-1 on the mycelia of *Fusarium* spp., the culture filtrate obtained under different fermentation times (1–10 d) was assessed. It was found that the culture filtrate after 1 d of culture had no significant inhibitory effect on *Fusarium* growth. The cell-free culture filtrate after fermentation for 2 d, however, had a very significant inhibitory effect on fungal growth; the percentage inhibition exhibited by the filtrate of cultures grown for 3–10 days was basically unchanged, reaching a plateau from day 3 (Appendix A). Thus, the cell-free culture filtrate after 3 d fermentation was used for subsequent tests. In addition, we analyzed the concentration dependence of the inhibitory effect of strain LY-1 on fungal growth using test concentrations of the culture filtrate of 0%, 5%, 10%, and 20% and found that, with the increasing concentration of the LY-1 culture filtrate, the percentage inhibition tended to increase (Appendix A). Therefore, we subsequently used the cell-free filtrate of 3-d cultures of LY-1 incorporated into potato dextrose agar (PDA) at a proportion of 20% to test the fungistatic effect of the metabolites of LY-1 on *Fusarium* spp. (Figure 2A,B). The results showed that, compared with the control, the growth inhibition values of *F. oxysporum*, *F. proliferatum,* and *F. solani* were 79.19%, 74.38%, and 83.80%, respectively (Figure 2C). This demonstrated that the LY-1 metabolites in cell-free culture filtrates could significantly inhibit the growth of the peanut root rot pathogen *Fusarium* spp., results that were similar to the fungistatic effects of the LY-1 bacterial suspension.

On the other hand, the percentage inhibition of fungal growth in response to the volatile organic compounds (VOCs) generated by strain LY-1 cultures was relatively weak (Figure 2A,B), with the percentage inhibition of *F. oxysporum*, *F. proliferatum*, and *F. solani* growth inhibition being 26.01%, 22.50%, and 11.26%, respectively (Figure 2C).

### 2.3. Inhibition of Sporulation and Conidium Germination of three Fusarium spp. by B. subtilis LY-1 Culture Filtrate

We tested the effect of the culture filtrate of LY-1 incorporated at a concentration of 20% into the PDA growth medium on the sporulation of three *Fusarium* spp. The results showed that the inhibitory effect of LY-1 culture filtrate on sporulation of the three *Fusarium* spp. was as follows: *F. solani* > *F. oxysporum* > *F. proliferatum*, with the percentage inhibition of each species being greater than 72.38% (Figure 3A). Similarly, the percentage inhibition of LY-1 culture filtrate on the germination of conidia of the three *Fusarium* species was also significant, with inhibition of at least 62.37% (Figure 3B). It is worth noting that the percentage inhibition of LY-1 on both sporulation and germination of *F. solani* was greater than 95% (Figure 3).

### 2.4. Production of Extracellular Enzymes by B. subtilis LY-1 

The presence of extracellular enzyme activity produced by strain LY-1 was measured using indicator media. The results showed that there were obvious transparent circles around the LY-1 colony on the three indicator media of protease, cellulase, and amylase (Figure 4), indicating that *B. subtilis* LY-1 could secrete these three enzymes, giving it the ability to decompose protein, cellulose, and starch. In addition, chitinase activity and β-1,3-glucanase activity in culture filtrate were quantitatively analyzed. The results showed that both chitinase and β-1,3-glucanase activity were present in the culture filtrate of *B. subtilis* LY-1, but that β-1,3 glucanase activity was greater than chitinase activity (Table 1).

### 2.5. Amplification of Genes Encoding Antifungal Peptides 

For the detection of genes responsible for the synthesis of antimicrobial peptides, five genes (iturin A synthetase C, *ituC*; bacillomycin L synthetase B, *bmyB*; bacilysin biosynthesis protein, *BacA*; surfactin synthetase subunit 1, *srfA*; and fengycin synthetase, *fenD*) were amplified by PCR using specific primers (Appendix A). The results showed that *ituC* and *bmyB* could be detected as a single target band, while the other three genes were not amplified to the target band (Appendix A). The bacillomycin synthesized by *bmyB* was an isomer of iturin and belonged to the iturin family. Therefore, *B. subtilis* LY-1 was capable of synthesizing the antifungal peptide iturin.

### 2.6. Growth-Promoting and Biocontrol Activity of B. subtilis LY-1 in Peanut Seedlings

The fresh and dry weights of peanut seedlings irrigated with culture filtrate of *B. subtilis* LY-1 were significantly greater than those of the control seedlings grown with only LB medium (Figure 5A) by 30.77% and 27.27%, respectively (Figure 5B).

In addition, co-inoculation of peanut seedlings with *B. subtilis* LY-1 and *F. oxysporum* resulted in a significant decrease in the disease index of peanut root rot compared with the control seedlings inoculated with only *F. oxysporum*; the disease index of the treatment and control groups was 70.83 and 39.17, respectively (Figure 5C). The biocontrol efficiency was as high as 44.71%, indicating that *B. subtilis* LY-1 could significantly inhibit the growth of *F. oxysporum* and reduce the incidence of peanut root rot.

## 3. Discussion

In recent years, biological control has become widely recognized as an important tool for the management of phytopathogens at the field scale. As biocontrol agents, antagonistic microorganisms have greater environmental friendliness and sustainability than conventional fungicides, and the identification and application of effective antagonistic biocontrol microorganisms have become a research hotspot. In this study, the *B. subtilis* strain LY-1, isolated from the rhizosphere of wild mint (*Mentha haplocalyx* Briq.), was demonstrated to achieve the effect of biological control on *Fusarium* spp. as well as directly promote peanut plant growth. 

Some members of the *Bacillus* genus have been reported to exhibit strong biocontrol activity against various plant pathogens. For example, Abdelmoteleb et al. reported that three strains of *B. subtilis* (LDA-1, 2, and 3) showed highly antagonistic efficacy against phytopathogenic fungi, with percentage inhibition ranging from 43.3% to 83.5% [20]. In this regard, the results of the dual-culture assay showed that *B. subtilis* strain LY-1 has potent antifungal activity against *F. oxysporum*, *F. proliferatum*, and *F. solani*, achieving percentage inhibition values of greater than 73.13%, and this effect was higher and more stable than that of other strains previously reported [20,21]. Conidia play a key role in the *Fusarium* disease cycle, so rapid production of conidia as a source of inoculum can lead to a higher subsequent infection rate and disease development [4]. In light of this, *B. subtilis* LY-1 significantly reduced the sporulation of each of the three *Fusarium* spp., as well as inhibiting the germination of the resulting conidia. LY-1 not only inhibited the mycelial growth of *Fusarium* spp., but also reduced the number of functional conidia. Through the comprehensive inhibitory mechanisms against *Fusarium* achieved by *B. subtilis* LY-1, the percentage infection of peanuts by *Fusarium* was significantly decreased, lowering the disease index of Fusarium root rot in peanuts and increasing the disease control effect.

The potential of *Bacillus* species to inhibit plant pathogenic fungi may be due to the secretion of antifungal molecules. The antagonistic activity of *Bacillus* against such fungi is mainly reflected in the synthesis of various antifungal peptides [22,23], secreted enzymes [24], such as chitinases and β-1,3-glucanases [25], other proteins [26], and volatile organic compounds (VOCs) [27]. *B. subtilis* LY-1 was shown to produce cellulase, protease, and amylase activity. Cellulases are important antagonistic factors for most biocontrol strains. Studies have shown that cellulase produced by *Trichoderma viride* plays an important role in inhibiting the mycelial growth of *Pythium aphanidermatum* [28]. On the other hand, 10% of the fungal cell wall is composed of proteins and glycoproteins. Proteases can degrade the cell wall, deform mycelia, etc. It has also been suggested that amylases contribute to the rhizobacterial colonization of roots, thereby possibly playing an important role in plant growth stimulation [29]. In addition, we detected chitinase and β-1,3-glucanase activities in cell-free culture filtrate of *B. subtilis* LY-1, which could degrade the cell wall of pathogenic fungi and negatively affect mycelial growth. We also amplified the *ituC* and *bmyB* genes for the synthesis of the antimicrobial polypeptide iturin. It has been reported that iturins exhibit strong fungal toxicity by forming ion-conduction pores upon contact with fungal cell membranes, damaging the plasma membrane, and causing leakage of cell contents [30]. Furthermore, with *F. graminearum*, iturin can cause morphological distortion of conidia and hyphae [31]. The biocontrol mechanism of *B. subtilis* LY-1 was analyzed by preliminary determination of antagonistic factors, but the main antagonistic substances and biocontrol mechanism still need to be identified and investigated.

In addition, our results also showed that *B. subtilis* LY-1 decreased the incidence of root rot disease in peanut seedlings. When the roots were irrigated with culture filtrate of *B. subtilis* LY-1, the control efficacy reached 44.71%. Several previously published potential biocontrol agents have been reported to exhibit similar or higher efficacies. For example, *Bacillus velezensis* (SM-39) and *Bacillus cabrialesii* (SM-93) significantly suppressed *Fusarium* root rot severity in wheat (by 42–62%) [32], whereas the biocontrol efficacy of *B. velezensis* strain B6 against *Fusarium* wilt in cabbage seedlings was reported to be 55.18% [6]. The biological control of *Fusarium* root rot in tomatoes by *B. velezensis* SDTB038 was 42.98% [33]. In view of the above reports, the differences in the level of biological control of *Fusarium* species among different biocontrol bacteria may also be related to the ability of the rhizobacteria to colonize the crops. The biocontrol effect of *B. subtilis* LY-1 is within a similar range to that of other published reports, and it has good application prospects, combining both direct growth promotion and broad-spectrum biocontrol activities (against three different *Fusarium* spp.). 

Many microorganisms have the capacity to promote plant growth, and *Bacillus* species are a major type of plant growth-promoting rhizobacteria (PGPR) [34]. PGPRs can indirectly promote plant growth by controlling phytopathogens by inducing systemic resistance, antibiosis, or competitive niches [35]. A direct way to promote plant growth is by producing compounds that directly stimulate plant growth or ameliorate stresses other than the plant pathogen in question [36]. In the current study on the growth-promoting effects of *B. subtilis* LY-1, both the fresh and dry weights of peanut plants were significantly higher than those of control plants. It is speculated that there may be a series of genes or gene clusters related to plant growth promotion in the strain LY-1 genome, including the synthesis of iron carriers, the production of growth-promoting volatile organic compounds or growth-promoting hormones, and mechanisms to achieve improved nutrient utilization, and these strategies need further investigation and verification. 

In conclusion, the isolation, identification, and application of *B. subtilis* LY-1 described in this study provide high-quality resources for further research and development into this biocontrol agent. The culture conditions of *B. subtilis* LY-1 can be further optimized, and the main antifungal molecules in the culture filtrate, including possible elicitors of induced systemic resistance, need to be explored and identified. 

## 4. Materials and Methods

### 4.1. Isolation of Antagonistic Bacillus subtilis LY-1 from Wild Mint Rhizosphere

The bacterial strain *Bacillus subtilis* LY-1 used in this study was isolated from wild mint (*Mentha haplocalyx* Briq.) collected in the Olympic Sculpture Park at Qingdao, Shandong Province, China. A sample (10 g) of rhizosphere soil was added to 90 mL distilled water and shaken for 15 min at 150 rpm. After allowing the suspension to settle, the supernatant was diluted by the 10-fold series method, and four concentrations (10^−2^ to 10^−5^) were used. An aliquot (150 μL) of each dilution was spread on LA (Luria-Bertani-Agar) plates and incubated at 25 °C for 24 h. Single colonies were selected, and streak purification was performed on LA plates. Three pure cultures were maintained at 4 °C on LA slants for further studies. The strains were stored in LB (Luria-Bertani-Broth) containing 25% glycerol (*v*/*v*) at –70 °C. 

### 4.2. Molecular Identification of Isolate LY-1

A single colony of isolate LY-1 was inoculated in a 10 mL Eppendorf tube containing 5 mL of LB broth medium and cultured at 25 °C for 18 h. Then, the DNA was extracted from the bacterial suspension with a bacterial DNA extraction kit (TIANamp Bacteria DNA Kit, TIANGEN Biotech, Beijing, China) by centrifugation at 10,000× *g* for 5 min at 4 °C, and the DNA was stored at −20 °C prior to use. The 16S rRNA and DNA gyrase subunit A (*gyrA*) genes were amplified to identify strain LY-1 [37,38]. PCR analysis was performed according to an established protocol [39]. PCR products were sent to Beijing Qingke Biotechnology Co., Ltd. (Beijing, China) for sequencing. The sequences obtained were submitted to GenBank database (https://blast.ncbi.nlm.nih.gov), and they were analyzed using the BLAST search, with 16S rDNA and *gyrA* sequences being compared with the most similar species. Based on these sequences, MEGA6.0 software was used to construct the neighbor-joining phylogenetic tree.

### 4.3. Test Strain and Culture Condition

The isolates used in this study of *Fusarium oxysporum*, *Fusarium proliferatum*, and *Fusarium solani* were preserved in our laboratory. The fungi were cultivated on PDA (potato dextrose agar) medium. The plates were incubated at 25 °C. *Bacillus subtilis* LY-1 was isolated in our laboratory and stored at the General Microbiology Center of the China Microbiological Culture Preservation Management Committee (CGMCC No.23930).

### 4.4. In Vitro Antagonistic Activity Assay 

To prepare a bacterial suspension, a single colony of strain LY-1 was selected and incubated in 20 mL of liquid LB at 25 °C at 150 rpm for 14 h. The suspension was centrifuged at 5000× *g* for 10 min. The supernatant was discarded, and the pellet was resuspended in sterile water to obtain a bacterial suspension with an OD_600_ of 0.2. The pathogens causing peanut root rot (*F. oxysporum*, *F. proliferatum* and *F. solani*) were inoculated into the center of PDA plates and incubated at 25 °C for 7 days in the dark. In vitro dual-culture analysis involved inoculating pathogen plugs with a diameter of 5 mm onto the center of a PDA plate, and 10 µL aliquots of LY-1 bacterial suspension at approximately 22 mm away from the fungal plug on four sides; 10 µL aliquots of sterile water were used as the control. Each treatment was replicated three times and incubated at 25 °C for 5 days. The colony diameter was measured, and the percentage inhibition of fungal growth was calculated. 

### 4.5. Antagonistic Activity of Extracellular Bacterial Metabolites 

The bacterial suspension was shaken for 48 h, the supernatant was collected by centrifugation, and the cell-free culture filtrate was obtained by filtration of the supernatant through a 0.22-µm filter membrane (Biosharp, Beijing, China). The cell-free culture filtrate was mixed at a 20% concentration with PDA culture medium that had been autoclaved, then cooled to about 50 °C, and the culture filtrate medium was poured into plates; control plates contained the same volume of water as the culture filtrate. A mycelial plug 5 mm in diameter of each of the above pathogens was placed in the center of each plate; each treatment was replicated three times, and the colony diameter was measured after incubation at 25 °C for 5 days to calculate the percentage inhibition.

### 4.6. Antagonistic Activity of Bacterial Volatile Organic Compounds

The antagonistic activity of volatile organic compounds was determined by the double-dish method [40]. An aliquot (100 µL) of the bacterial suspension (OD_600_ = 0.2) was spread on LB plates, and the 5-mm diameter fungal plug was placed in the center of each PDA plate. Each treatment was repeated three times, the colony diameter was measured, and the percentage fungal growth inhibition was calculated at 25 °C for 5 days.

### 4.7. Production of Extracellular Enzymes 

The indicator media (for enzymes protease [41], amylase [42], and cellulase [43]) were inoculated with 10 μL LY-1 bacterial suspension in the center of the plate and cultured at 28 °C to observe whether there was a transparent circle.

Chitinase kit (JDZM-2-G, Comin, Suzhou, China) and β-1,3 glucanase kit (GA-2-Y, Comin, Jiangsu, China) were used to quantify extracellular enzyme activity in the cell-free culture filtrate of *B. subtilis* LY-1. The method for the chitinase assay is as follows: Control (CK): sterile culture filtrate (400 µL), extract solution (600 µL); Treatment (T): sterile culture filtrate (400 µL), extract solution (200 µL), and reagent (1, chitin solution) (400 µL). The reaction solutions were mixed well and incubated in a water bath at 37 °C for 1 h. The reaction was terminated by incubation in a boiling water bath for 5 min, then cooled at 4 °C, centrifuged at 8000× *g* for 10 min, and the supernatant collected and diluted with distilled water 10-fold. Then, 700 µL supernatant was added to the reagent (2, 3,5- dinitrosalicylic acid solution) (500 µL), mixed well, incubated at 95 °C for 5 min, and cooled to room temperature. An aliquot (1000 μL) was absorbed onto 1 mL glass colorimetric plates, and the light absorption values A (T) and A (CK) at 540 nm were determined using a UV spectrophotometer (UV-5800H, Shanghai Yuanxi Instrument Co., Ltd., Shanghai, China), ΔA = A (T) − A (CK).
Chitinase activity (mg/h/mL) = (ΔA + 0.2753) ÷ 6.4108 × V (total volume of reaction mixture, 1 mL) ÷ V (the sample volume in the reaction mixture, 0.4 mL) × 10 (dilution ratio) = 3.899 × (ΔA + 0.2753)

To quantify β-1,3 glucanase activity, the following reagents were successively added to the 1.5 mL Eppendorf tube. CK (sterile culture filtrate 100 µL and ddH_2_O 100 µL), Treatment (T) (sterile culture filtrate 100 µL and reagent (1, laminarin) 100 µL). The reagents were thoroughly mixed and incubated in a 37 °C water bath for 60 min. Reagent (2, 3,5- dinitrosalicylic acid solution) (600 µL) was added to CK and T, thoroughly mixed, incubated at 95 °C for 5 min (cover tightly to prevent water loss), then cooled and the A_550_ value of each tube determined, ΔA = A(T)–A(CK).
β-1,3-Glucanase (mg/h/mL) = [(ΔA + 0.0192) ÷ 0.0958 × V1] ÷ V1 = 10.438 × (ΔA + 0.0192)

V1: Sample volume added to the reaction system, = 0.1 mL. 

### 4.8. Amplification of Antimicrobial Peptides Genes

Genes of antimicrobial peptides (surfactin, bacilysin, fengycin, bacillomycin, and iturin) were identified by PCR amplification using specific primers (Appendix A) [21].

### 4.9. Effects of B. subtilis Strain LY-1 on the Growth and Disease Level of Peanut 

Healthy 2-week-old peanut seedlings were selected for growth promotion and biocontrol studies, with three seedlings per pot, for a total of 24 pots (top diameter 20 cm, bottom diameter 16 cm, and height 14 cm). The substrate used for growing peanuts is sandy loam dug directly from peanut fields, and the clay composition is about 40%. 12 pots of the seedlings were irrigated with 10 mL of LY-1 culture filtrate (treatment, T) per pot, and the other 12 pots of seedlings were irrigated with 10 mL of LB medium (control, CK) per pot, each diluted to 60 mL with LB medium. After 2 weeks, the control and treatment were divided into two parts. One sample (6 pots of seedlings were irrigated with LY-1 culture filtrate and 6 pots of seedlings were irrigated with LB medium) was harvested, and the fresh and dry weights of the plants were measured (drying at 105 °C for 30 min, then drying at 75 °C to a constant weight). In the other sample (the other 6 pots of seedlings were irrigated with LY-1 culture filtrate, and the other 6 pots of seedlings were irrigated with LB medium), each peanut plant was inoculated with 10 oat grains covered with mycelium of *F. oxysporum*. After 2 weeks of infection, the disease index and percentage inhibition effect were calculated. The experiment was repeated twice.

Grading standards: Level 0, healthy, asymptomatic; Level 1, rot occurs only on the root surface, rot area ≤ 10%, or rot has been extended to internal and decaying area ≤ 10%; Level 2, the decay has been extended to the interior, the decay area is 10% to 40%, and the fibrous roots are not rotten. Level 3, the decay has been extended to the interior, the decay area is 40% to 50%, and the fibrous roots are rotten. Level 4, the decay has been extended to the interior, the decay area is 50% to 70%, and the fibrous roots have decayed and fallen off. Level 5, the rot has spread to the interior and the rot area is >70% until the whole root is completely rotted, the fibrous root rots, or the whole plant wilts, lies down, or dies.

### 4.10. Statistical and Data Analyses 

Microsoft Excel (2016) was used for statistical analysis, and Canvas11 software was used for making figures in this article. A one-way analysis of variance with multiple comparisons using Duncan’s multiple range test was used for the analysis.
Antibacterial percentage (%) = (C − T)/C × 100,
where C = control colony growth diameter and T = treatment colony growth diameter.
Disease index = ∑ (the value of each level × the number of diseases at each level)/(the highest level × the total number of investigated plants) × 100
where the value of each level (e.g., 0,1,2,3,4,5), the highest level (=5), and the total number of investigated plants (=18)
Inhibition effect (%) = (control disease index − treatment disease index)/control disease index × 100 

## 5. Conclusions

An effective biocontrol strain, LY-1, was isolated from rhizosphere soil samples of wild mint by agar disk dilution and dual-culture assay, and it was identified as *Bacillus subtilis* based on morphological characteristics, and 16S rDNA and *gyrA* sequence analysis. In this study, it was found that *B. subtilis* LY-1 had strong fungistatic activity against mycelium growth, conidial production, and conidial germination of three *Fusarium* spp. which cause peanut root rot. The mechanism of the antifungal activity depends on the fact that *B. subtilis* LY-1 can not only produce protease, cellulase, and amylase but also chitinase and β-1,3-glucanase activities and the antifungal lipopeptide, iturin. At the same time, LY-1 can directly promote the growth of peanut seedlings and effectively prevent and control the occurrence of peanut root rot. Finally, *B. subtilis* LY-1 has good prospects as a biocontrol agent, with growth-promoting and fungistatic activity.

## Figures and Tables

**Figure 1 plants-12-04158-f001:**
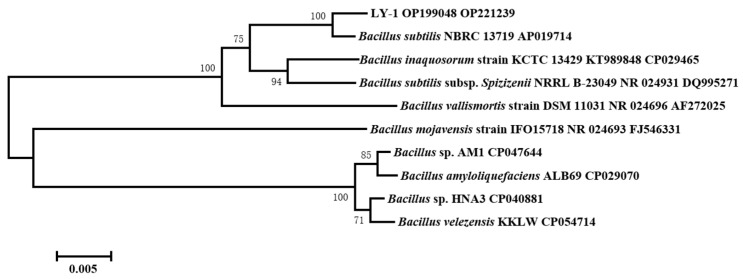
Phylogenetic tree of LY-1, based on the sequences of 16S rDNA and *gyrA* genes.

**Figure 2 plants-12-04158-f002:**
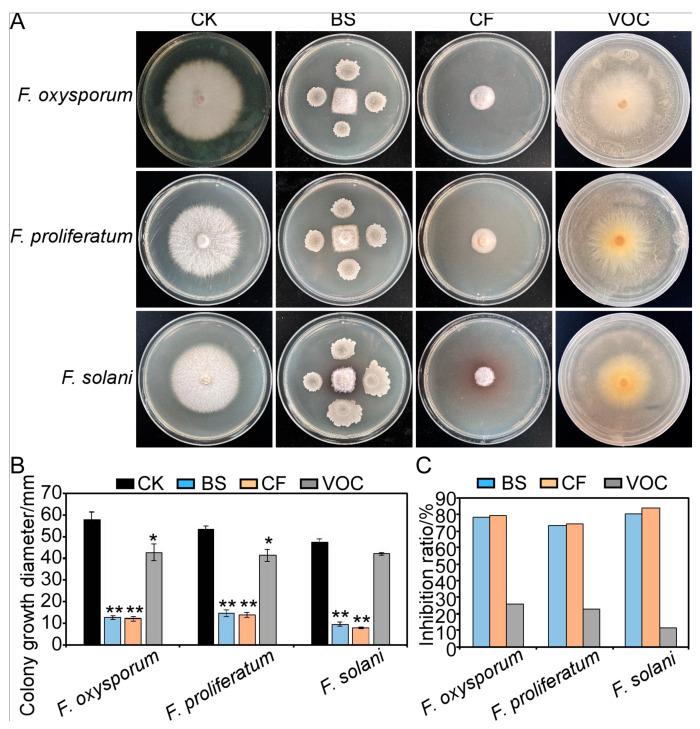
Antagonism of *Bacillus subtilis* LY-1 against *Fusarium* spp. (**A**) Fungistatic effect of LY-1 on three *Fusarium* spp.; the fungal colonies were photographed on day five. (**B**) Analysis of colony growth of pathogens (error bars = SE). (**C**) Analysis of percentage inhibition by LY-1 of growth of three *Fusarium* spp. BS: dual culture of fungal colonies (center), replicate bacterial suspension (periphery); CF: cell-free culture filtrate incorporated in potato dextrose agar (PD); VOC: volatile organic compounds. CK refers to the control, showing growth of *Fusarium* spp. colonies on the PDA medium without any treatment. ** indicates significant difference from the CK treatment at the *p* < 0.01 level; * indicates significant difference from the CK treatment at the *p* < 0.05 level.

**Figure 3 plants-12-04158-f003:**
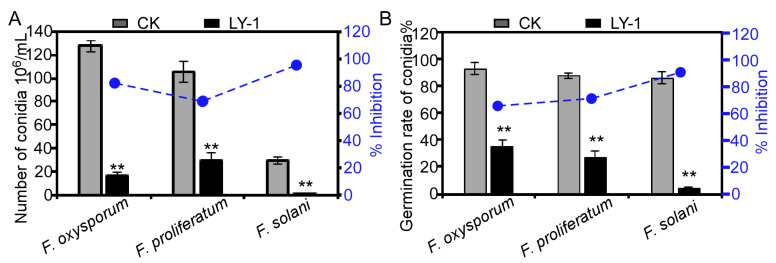
Effect of 20% culture filtrate of *Bacillus subtilis* LY-1 on sporulation and conidium germination of three *Fusarium* species. (**A**) Bars indicate number of conidia produced by each species, and the blue dot shows the percentage inhibition by LY-1 culture filtrate on sporulation relative to the control (CK). (**B**) Bars indicate the percentage germination of conidia produced by each species, and the blue dot shows the percentage inhibition by LY-1 culture filtrate on percentage germination relative to the control (CK). ** indicates significant difference between control (CK) and LY-1 treatment at the *p* < 0.01 level; error bars indicate SE.

**Figure 4 plants-12-04158-f004:**
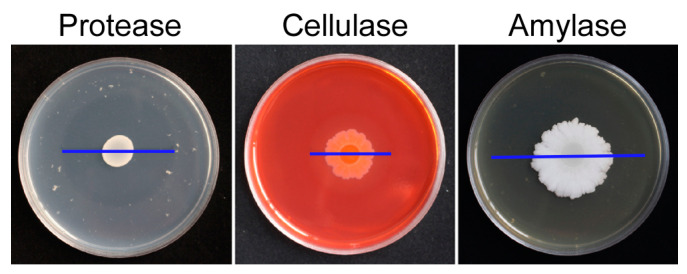
Detection of *B. subtilis* LY-1 extracellular enzyme activity. Note: The blue line represents the diameter of the transparent circle.

**Figure 5 plants-12-04158-f005:**
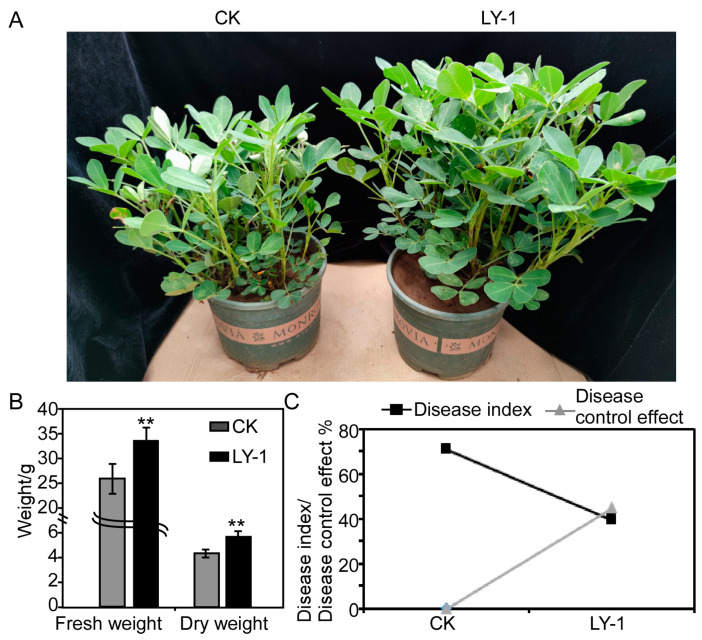
Analysis of growth-promoting and disease-control effects of LY-1 inoculation on peanut seedlings and *Fusarium* root rot. (**A**) Growth promotion of peanuts by LY-1 culture filtrate. (**B**) Fresh mass and dry mass of peanut seedlings treated with LY-1 culture filtrate. (**C**) *Fusarium* root rot disease index and disease control effect of *Bacillus subtilis* and *Fusarium oxysporum* (LY-1) co-inoculation or *F. oxysporum* only (CK) on peanut seedlings. ** indicates significant difference between control (CK) and LY-1 treatment at the *p* < 0.01 level; error bars indicate SE.

**Table 1 plants-12-04158-t001:** The activity of chitinase and β-1,3-glucanase in cell-free culture filtrate of *B. subtilis* LY-1.

Degrading Enzymes	A(C)	A(T)	ΔA	Activity (mg/h/mL)
Chitinase	0.064 ± 0.002	0.088 ± 0.002	0.024	1.167
β-1,3-glucanase	0.036 ± 0.001	0.297 ± 0.001	0.261	2.925

## Data Availability

The corresponding author can provide the data backing up these conclusions upon reasonable request.

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
