# Peer review of "Isolation and Identification of Bacillus subtilis LY-1 and Its Antifungal and Growth-Promoting Effects"

_plants, 2023, doi:10.3390/plants12244158_

Round 1

Reviewer 1 Report

Comments and Suggestions for Authors

The authors are researching finding an ecological solution to treat the blight caused by the species Fusarium sp. on peanut roots. In this case, I use a biological control microorganism identified as the Bacillus subtilis bacterium, which cultivated and analyzed in the laboratory led to promising results, on the 6 species of Fusarium tested. Investigation techniques appropriate to the proposed objective were used. Of course, the research should be continued outside the laboratory where there are more factors that interact.

Author Response

Thank you very much for your comments and support. We will conduct further tests on the biocontrol effect of Bacillus subtilis LY-1 in the disease field. 

Reviewer 2 Report

Comments and Suggestions for Authors

The manuscript submitted for review entitled. "Isolation and Identification of Bacillus subtilis LY-1 and Its An-tifungal and Growth-promoting Effects" is an interesting study with cognitive and utilitarian values. The authors have presented laboratory studies which, with sufficient repetition, provide a basis for inference. I strongly request that the authors take into account the comments below.

2. Results

2.1. Isolation and identification of a Bacillus subtilis strain LY-1

- Paragraph - „The 16S rDNA and gyrA gene fragments of LY-1 were obtained by PCR amplification. The effective length of the two gene sequences obtained were 1391 bp and 783 bp, respectively. The sequences were submitted to GenBank data-base (OP199048 and OP221239) for BLAST analysis and comparison. Based on 16S rDNA and gyrA gene sequence, MEGA6.0 software was used to construct phylogenetic tree”. – should be transferred to 4. Materials and Methods

2.2. Antagonistic activity of B. subtilis LY-1 against peanut Fusarium spp.

- in the sentences - „As shown in Figure 2, the strain LY-1 demonstrated antagonistic activity against Fusarium spp., namely F. oxysporum, F. proliferatum and F. solani in dual cultures compared to the control plates. The bacteria suspension of strain LY-1 significantly suppressed the hyphal growth of these three fungal species Fusarium spp., and the inhibition rates were 78.61%, 73.13% and 80.28%, respectively (Figure 2C). – remove deleted words, insert in red font.

- in Figure 2. A – it is necessary to write on which day of rearing the fungal colonies were photographed. No explanation CK – add.

-from the sentence „This revealed that the metabolite of LY-1 fermentation broth can sig-nificantly inhibit the growth of peanut root rot pathogen Fusarium spp. – but at a level not significantly different from BS (Figures 2. B, C). The strength of the fungistatic effect of FSB and BS is comparable - note this.

- the description of Figure 3 should include information that it is a 20% fermentation solution of B. subtilis LY-1.

3. Discussion

- page 7- „Through the comprehensive inhibition of Fusarium by B. subtilis LY-1, the infection rate of Fusarium to peanut was reduced, thereby the disease index of peanut root rot was reduced and the control effect was improved, thus increasing the peanut yield.” - Can increase yield! peanut yield was not assessed in these studies.

4. Materials and Methods

Explain all abbreviations of culture media used and specify their composition (LA, LB, PDA and selective media included in 4.7.).

-  4.1. - Why were B.subtilis bacteria isolated from wild mint? Be sure to explain the abbreviation LA!

- 4.3. – Were the fungi: Fusarium oxysporum, Fusarium proliferatum and Fusarium solani were isolated from diseased peanut roots?

-4.4. Do three repetitions in laboratory tests give rise to a reliable conclusion?

-4.5. In the sentence „According to the method in 2.4,...” - should be 4.2.

-4.8. The motodology contained in this subsection must be written again. It should be described in detail, what was the size of the vases, what was the substrate, why only F.oxysporum and not F. proliferatum and F.solani were used for inoculation, when were the Ly-1 treatments performed - all described in detail! Finally, how many combinations there were and how many repeats - this is not clear! Please state how the plant health assessment was done, on how many plants it was done. Since a disease index is given in 4.9., please very much describe the infestation scales, what was the highest scale level and what was the lowest scale level and what did they mean?

5. Conclusions

It is not the best written, more specificity and clarification would be useful. Are you sure the authors are referring to an "antimicrobial" effect . The tests were carried out on fungi so it is about fungistatic activity.

Reviewer 3 Report

Comments and Suggestions for Authors

This research highlights the biocontrol potential of a B. subtilis isolate against Fusarium spp. The main concern with this research is novelty. This is almost more than a couple of decade old phenomenon, and there are numerous such reports, with more details of genomic attributes, mechanisms, interactions, expressions etc. This work does not stand to the merits of novelty and does not provide any insight into the biocontrol process. This needs to be addressed. Other concerns are given below:

1.       Don’t italicize ‘spp.’ Or ‘sp.’ after the genus.

2.       “…..identified a new antagonistic Bacillus subtilis strains isolated from..” correct ‘strain’.

3.       Revise / correct the sentences in section 2.1.

4.       Section 2.2, bacterial suspension

5.       ‘Sterilized fermentation broth’ is not a very appropriate term. Is it ‘cell-free culture filtrate’ instead?

6.       According to the statistical analysis, compared with the control….. rephrase the sentence.

7.       Its not growth inhibition ‘rate’, but ‘percentage’.

8.       Section 4.5, correct the first sentence.

9.       There is no information on the active constituent in ‘volatile gas’ and ‘sterile culture filtrate’, that resulted in mycelial inhibition. This is required to understand the biocontrol mechanisms. Also, its needed as part of the ‘characterization’ of said isolate, as highlighted in the title.

Comments on the Quality of English Language

Language need to be revised and corrected.

Reviewer 4 Report

Comments and Suggestions for Authors

Comments on the Quality of English Language

--

Round 2

Reviewer 3 Report

Comments and Suggestions for Authors

The authors have done significant efforts with the addition of some relevant information in revised manuscript.

Comments on the Quality of English Language

its okay